# Spatial Distribution, Antioxidant Capacity, and Spore Germination-Promoting Effect of Bibenzyls from *Marchantia polymorpha*

**DOI:** 10.3390/antiox11112157

**Published:** 2022-10-31

**Authors:** Jiao-Zhen Zhang, Chan Wang, Ting-Ting Zhu, Jie Fu, Hui Tan, Cheng-Min Zhang, Ai-Xia Cheng, Hong-Xiang Lou

**Affiliations:** Key Laboratory of Chemical Biology of Natural Products (Ministry of Education), School of Pharmaceutical Sciences, College of Medicine, Shandong University, Jinan 250012, China

**Keywords:** *Marchantia polymorpha*, bibenzyls, UV-B radiation, antioxidant ability, spore germination

## Abstract

Liverworts, considered to be the first plant type to successfully make the transition from water to land, can resist different oxidative stress. As characteristic constituents of liverworts, the bibenzyls are efficient antioxidants. In this study, spatial distributions of the bibenzyls within *Marchantia polymorpha* L., the model species of liverworts, were mapped using airflow-assisted desorption electrospray ionization imaging mass spectrometry. Bibenzyls were found to largely exist in the female receptacle of *M. polymorph**a*, where lunularic acid was found to focus in the central region and bisbibenzyls were enriched in the periphery. The region-specific gene expression and antioxidant activities were characterized. In line with the spatial feature of bibenzyls, higher MpSTCS1A and Mp4CL expression levels and antioxidant ability were exhibited in the archegoniophore. The expression level of MpSTCS1A, and the content of total phenolic acid was increased after UV-B irradiation, suggesting bibenzyls play an important role in UV-B tolerance. Moreover, lunularic acid and extract of archegoniophore at a certain concentration can stimulate the spore germination under normal conditions and UV-B stress. These works broaden our understanding of the significance of bibenzyls in spore propagation and environmental adaptation.

## 1. Introduction

Known as pioneer plants transitioning to terrestrial habitats [1,2], liverworts produce plenty of polyphenols, devoted to terrestrial physiological adaptation such as UV radiation tolerance [3,4]. Distinguished from flavonoids rich in other higher land plants, bibenzyls are characteristic polyphenols of liverworts, which are synthesized involving stilbenecarboxylate synthase (STCS1) and 4-coumarate: coenzyme A ligase (4CL), and exhibit antioxidant potential as free radical scavengers [5,6].

Located basally in land plant genealogies, and short on vascular tissues, true roots and flowers, liverworts are gametophyte dominant plants, with occasionally appearing sporophytes, morphologically and nutritionally relying on the gametophytes [7]. *Marchantia polymorpha* L., the representative species of dioecious liverworts, can develop into female and male gametophytes, which differentiate into archegoniophore and antheridiophore, respectively [8]. The reduced sporophytes are sheltered by the archegoniophore for their development. Therefore, spatial profiling of metabolites from female and male *M. polymorpha* could be conducive to obtaining further insights into spore propagative protection and environmental adaptation of liverworts.

Among the modern techniques employed for plant metabolomics, mass spectrometry imaging (MSI) has developed as a powerful approach for microscopically visualizing the molecular distributions of plant tissue, in an untargeted and label-free way [9,10]. In this study, with the aid of one direct and sensitive ionization method: airflow-assisted desorption electrospray ionization (AFADESI) MSI [11,12], the spatial distribution of metabolites in *M. polymorpha* at the reproductive stage has been visualized for the first time. Supported by high mass accuracy and in situ tandem MS (MS/MS), various bibenzyls were identified as predominate constituents, which were further observed in the archegoniophore and thallus of female *M. polymorpha* (ArM and TFM, respectively), rather than the antheridiophore and thallus of male *M. polymorpha* (AnM and TMM, respectively). The expression patterns of two key genes in the bibenzyl biosynthesis and the antioxidant capacity of the above four parts exhibited the same trend. In addition, the expression level of MpSTCS1A and the content of total phenolic acid in the thallus of *M. polymorpha* increased after UV-B radiation; meanwhile, bibenzyls were proven to be able to accelerate spore germination, suggesting the bibenzyls play an important role in spore propagative protection and stress adaptation as antioxidants. Herein, we report the spatial distribution of bibenzyls, analysis of gene expression patterns, evaluation of antioxidant potentials, adaptation to UV-B radiation, and spore germination rate in a gender and region-specific pattern.

## 2. Materials and Methods

### 2.1. Chemicals and Plant Samples

Formic acid (HPLC grade), acetonitrile (ACN; HPLC grade), and gelatin (gel strength ~250 g Bloom) were purchased from Rhawn (Shanghai, China), 3,4,5-Trihydroxybenzoic acid and Na_2_CO_3_ were obtained from Bide Pharmatech Co., Ltd. (Shanghai, China). Folin-Ciocalteu’s phenol reagent, 2,4,6-Tri(2-pyridyl)-S-triazine, 2,2′-azino-bis(3-ethylbenzothiazoline-6-sulfonate), K_2_S_2_O_8,_ and FeCl_3_·6H_2_O were purchased from Macklin (Shanghai, China).

The liverwort *M. polymorpha* was collected from Yandang Mountain, Wenzhou, China. The sterile thallus of *M.*
*poly**m**orpha* was obtained in the laboratory of Shandong University.

### 2.2. Sample Preparation of MSI and AFADESI-MSI Analysis

The tissues of fresh *M.*
*p**olymorpha* were embedded in gelatin solution (10% *w/v*) on the sample holder. The sample holder was placed on a −60 °C quick freeze table until it was completely solidified. The sample tissues were cryo-sectioned into 14 μm sections at −20 °C on cryostat microtome (Thermo CryoStar NX50 NOVPD, Bremen, Germany) and attached to an adhesion microscope slide (superfrost^TM^ plus slides, 25 × 75 × 1 mm, Epredia, Portsmouth, United States). A stereo microscope was used to obtain optical images of tissue sections. In addition, tissue sections were preserved in a −80 °C refrigerator for at least 24 h and were placed in a vacuum desiccator before being used for AFADESI mass spectrometry imaging.

The AFADESI-MSI system was coupled with a Q-Orbitrap mass spectrometer (Q Exactive, Thermo Scientific, Bremen, Germany) as described previously for MSI analysis [13]. The electrospray desorption solvent was an acetonitrile–water solution (*v/v* 4:1) and the flow rate was 5 µL/min. The spray voltage was set at −4500 V. The unidirectional scanning speed in the X direction was 100 or 200 µm/s and the inter-row distance in the Y direction was also 100 or 200 µm. Data in the 100–600 m/z range were acquired in negative full scan mode. Xcalibur Qual browser software (ThermoFisher Scientific, Waltham, MA, USA) was used to convert the format of the acquired data files from .raw to .cdf and Mass Imager was used for the processing and visualization of the data of MS spectra.

### 2.3. LC–MS/MS analysis

Fresh male and female *M. polymorpha* (100 mg, respectively) were extracted by methanol (10 mL) with ultrasound for 60 min; the mixture was then filtered through a 0.22 μm organic membrane before injection into an LC-MS system for analysis.

The Thermo Dionex Ultimate 3000 UHPLC system was used for chromatographic analysis. A C18 Hypersil Gold column (3 µm, 2.1 × 100 mm, Thermo Scientific) was utilized for separation at 35 °C with a flow rate of 0.3 mL/min. The mobile phase was 0.1% (*v/v*) formic acid (A) and methanol (B). The gradient program was performed as follows: 0–1 min, 25% B; 1–20 min, 25–95% B; 20–24 min, 95% B; 24–25 min, 95–25% B; 25–28 min, 25% B. The injection volume was 2 µL. MS was performed on a Q-Orbitrap mass spectrometer with an ESI source. The qualitative analysis was operated in Full MS/dd-MS^2^ (Top5) scan mode and scanned positive and negative ions together. The quantitative analysis was operated in negative ion full MS mode. The molecular weight scan range was 100–600. The ESI source parameters were as follows: sheath gas flow rate, 40; aux gas flow rate, 10; sweep gas flow rate, 0; spray voltage, 3.2 kV; capillary temperature, 300 °C; S-lens RF level, 50; aux gas heater temperature, 30 °C. Nitrogen was used throughout the process.

### 2.4. RT-Q-PCR Analysis

Total RNA from four tissues of *M. polymorpha* was extracted and purified using the cetyltrimethylammonium bromide (CTAB) method [14]. A Hiscript Q RT SuperMix for qPCR (+gDNA wiper) (Vazyme, Nanjing, China) was used in the reverse transcription reaction to convert RNA into cDNA following the manufacturer’s instructions. Aliquots of 1 µg of total RNA were used for each sample to synthesize first-strand cDNA during this process. The cDNA was diluted 5-fold before being used for RT-Q-PCR. The amplification reactions of RT-Q-PCR were performed on a Mastercycler ep gradient S 96 Real-Time PCR System (Eppendorf) with gene-specific primers, and the ChamQ SYBR qPCR Master Mix (Vazyme, Nanjing, China) was used to reflect differences in gene expression levels. The conditions of the thermal cycle were 2 min at 95 °C followed by 40 amplification cycles (15 s at 95 °C, 15 s at 53 °C, and 20 s at 68 °C). The primers used for RT-Q-PCR are shown in Appendix A. The expression patterns of key genes in the bibenzyl biosynthetic pathway were analyzed in three replicates.

### 2.5. Determination of Antioxidant Activity of Crude Extract in Different Tissues of M. polymorpha

#### 2.5.1. Preparation of Methanol Extracts

The ultrasonic extraction process of the different tissues of *M. polymorpha* was as described in Section 2.3 and 1 mg/mL crude extract solution was passed through a 0.22 μm organic filter again for analysis.

#### 2.5.2. Total Phenolic Acid Content (TPC) Determination

The content of total phenolic acid in the extracts was determined by the Folin-Ciocalteu phenol reagent method [15]. Briefly, 20 μL methanol extract, 100 μL Folin-Ciocalteu’s phenol reagent (*v/v* 1:10 in water), and 80 μL 7.5% Na_2_CO_3_ were added to a 96-well microplate in triplicate. An amount of 200 μL methanol was used as a blank solvent control. Then, the microplate was wrapped with aluminum foil and incubated in a 45 °C constant temperature incubator for 15 min. Absorbance values were measured at a wavelength of 765 nm in a microplate reader. A standard curve was prepared with different concentrations of gallic acid. The total phenolic acid content in the crude extract was measured by the standard curve (y = 0.0579*x* + 0.0085, R^2^ = 0.9991) and was expressed as mg of gallic acid equivalent per g of fresh weight (GAE mg/g FW). Values were presented as mean ± standard deviation of three replicates.

#### 2.5.3. DPPH Radical Scavenging

The antioxidant power of the extracts was evaluated by measuring the DPPH free radical scavenging ability and Trolox was used as standard [15]. A 15 μL methanol extract and 285 μL of a freshly prepared 60μM DPPH solution were added to a 96-well plate. An amount of 300 μL methanol was used as a blank solvent control. The microplate was covered with aluminum foil and allowed to react for 30 min at room temperature in the dark. The absorbance values against a blank were detected at a wavelength of 517 nm in a microplate reader. A standard curve was prepared with different concentrations of Trolox. The DPPH free radical scavenging ability of the crude extracts was measured by the linear standard curve (y = 0.0775*x* + 0.0173, R^2^ = 0.9945) and was expressed as μg of Trolox equivalent per g of fresh weight (TE μg/g FW). Values were presented as mean ± standard deviation of three replicates.

#### 2.5.4. ABTS Radical Scavenging

The diammonium 2,2′-azino-bis(3-ethylbenzothiazoline-6-sulfonate) (ABTS) radical scavenging activity was used to evaluate the antioxidant activity of the crude extract and Trolox was used as standard [15]. The ABTS stock solution was prepared by mixing an equal volume of 7 mM ABTS and 2.5 mM K_2_S_2_O_8_ and reacting at room temperature in the dark for 16 h. The ABTS stock solution was diluted 20-fold with a pH 7.4 PBS buffer to make the ABTS working solution before the experiment. An amount of 285 μL of freshly prepared ABTS working solution was mixed with 15 μL of methanol extract solution at room temperature and incubated in the dark for 10 min. An amount of 300 μL methanol was used as a blank solvent control. The absorbance relative to the blank was measured with a microplate reader at a wavelength of 734 nm. The ABTS free radical scavenging ability of the extracts was measured by the linear standard curve (y = 0.0957*x* − 0.0048, R^2^ = 0.9957) and was expressed as μg of Trolox equivalent per g of fresh weight (TE μg/g FW). Values were presented as mean ± standard deviation of three replicates.

#### 2.5.5. Ferric Reducing Antioxidant Power (FRAP)

The ferric reducing antioxidant power was often used as one of the indicators of antioxidation and Trolox was used as standard [15]. An amount of 10 mM 2,4,6-Tri(2-pyridyl)-S-triazine (TPTZ) solution dissolved in 40 mM HCl, 20 mM FeCl_3_·6H_2_O, and pH 3.6 acetate buffer was mixed at a volume ratio of 1:1:10 to prepare the Fe^3+^-TPTZ working solution. An amount of 25 μL of methanol extract and 275 μL of Fe^3+^-TPTZ working solution were added to a 96-well plate. After reacting at room temperature for 5 min, the absorbance was measured at a wavelength of 593 nm in a microplate reader. The ferric reducing antioxidant power of the extracts was measured by the linear standard curve (y = 0.1718*x* + 0.0155, R^2^ = 0.999) and was expressed as μg of Trolox equivalent per g of fresh weight (TE μg/g FW). Values were presented as mean ± standard deviation of three replicates.

### 2.6. Expression Patterns of STCS1s and TPC Response to UV Treatment

Seven-week-old thalli of male and female *M. polymorpha* were exposed to UV-B irradiation (*λ* = 302 nm) for 3 h, then put in an artificial climate incubator for normal cultivation (25 °C, 12 h/12 h). Samples were taken after 0, 6, 12, 24, 36, 48, and 60 h and immediately frozen in liquid nitrogen and stored at −80 °C. Sampling time points were counted after 3 h of exposure. The sample without UV treatment was taken as a control. Then, based on these materials, expression patterns of MpSTCS1A were analyzed by RT-Q-PCR. In addition, the same UV-treated fresh male and female leaves were taken, and the analysis of TPC was the same as those in Section 2.5.2 by the standard curve (y = 0.0483*x* + 0.0169, R^2^ = 0.9981).

### 2.7. Spore Germination Rate Assay

The spore germination rate was tested as described previously with some modifications [16]. The mature and unruptured sporangia were picked from female gametophyte, rinsed with distilled water, and suspended in water. The surface was disinfected with 75% ethanol for 5 min in the ultra-clean condition, rinsed three times with sterile water, and then sterile forceps were used to break the sporangia in an appropriate amount of sterile water to release the spores to make a spore suspension with an appropriate concentration liquid. Then, the spore suspension was spread on an agar medium (1% agar, 1/2 B5, 0.5‰ (*w*/*v*) MES, pH 5.7) containing various concentrations of LA and ArM extract. One group was cultured under normal conditions, and the other group was under UV stress for 30 min. The culture dish was placed in an artificial climate incubator for cultivation under 25 °C and a 12 h/12 h light–dark cycle. The spore germination of *M. polymorpha* was observed using an inverted microscope after culturing for three days.

### 2.8. Statistical Analysis

The data of the assays were all sorted in the GraphPad Prism software Version 8.00 for Mac (GraphPad Software, Inc., San Diego, CA, USA). At least three replicates were set for each set of data. Values were presented as mean ± standard deviation (SD). Means were compared by one-way analysis of variance (ANOVA) with Tukey’s test, and differences were considered significant at *p* < 0.05.

## 3. Results and Discussion

### 3.1. Data Mining of Characteristic Metabolites from M. polymorpha Using AFADESI

Metabolite profiling of four different parts from female and male *M. polymorpha* (ArM, TFM, AnM, and TMM, Figure 1A) was performed by combined analysis of LC-MS and AFADESI-MSI. Application of LC-MS/MS, coupled with Compound Discoverer software [17], preliminarily identified 292 metabolites (Appendix A), and the structures of 8 reference substances are shown in Appendix A. Bibenzyls were identified as major metabolite types, and negative-ion mode was selected as optimal for the following AFADESI-MSI. As shown in Figure 1B–D, spatial distributions of metabolites in different parts were presented by AFADESI-MSI through scanning the cross sections of ArM, TFM, AnM, and TMM in the negative mode. With the aid of the aforementioned qualitative analysis of LC-MS and AFADESI MS/MS results (Appendix A), the characteristic differential components are lunularic acid (LA, calcd. for [C_15_H_14_O_4_-H]^−^, 257.0816); isoriccardin C/isoriccardin D/marchantin C/isomarchantin C/neomarchantin A (IRC/IRD/MC/IMC/NMA, calcd. for [C_28_H_24_O_4_-H]^−^, 423.1600); and marchantin A (MA, calcd. for [C_28_H_24_O_5_-H]^−^, 439.1549). Multivariate statistical analysis was performed to quantify metabolite abundance from different parts of *M. polymorpha.* PCA analysis could screen part-specific discriminating components. As shown in Figure 1E, the score plot shows the good clustering between the female and male *M. polymorpha,* and it is obvious in the weighted loading plot that the ion at *m/z* 257.0816, 423.1600, and 439.1549 presented the discriminant ability of different parts.

### 3.2. Spatial Distribution of Bibenzyls in M. polymorpha

The spatial distributions of bibenzyls within *M. polymorpha* were mapped based on AFADESI-MSI. As shown in Figure 2, imaging data from the horizontal cross sections of ArM and AnM exhibited that compounds with m/z ions at 257.0816 (LA), 423.1600 (IRC/IRD/MC/IMC/NMA), and 439.1549 (MA) had stronger intensity in the female receptacle. Vertical cross sections of TFM and TMM showed a similar pattern of distribution. In addition, for ArM, LA was especially abundant in the central portion of archegoniophore where archegonia grow, and sporophytes develop, while other bisbibenzyls at m/z 423.1600 and 439.1549 are prominent in peripheral regions of fingerlike lobes, which indicated their protective effects against herbivores, microorganisms, and neighboring plants.

For further investigation of time- and space-specific distributions of bibenzyls, MSI analysis of ArM in different periods was performed. As shown in Figure 2, LA at m/z 257.0816 was discovered to be especially enriched in ArM with unruptured capsules (ArM-UC) compared to ArM with ruptured capsules (ArM-RC). However, the distribution of bisbibenzyls at *m/z* 423.1600 and 439.1549 were unable to display a significant difference between ArM-UC and ArM-RC. The MSI results suggested that compared to bisbibenzyls such as IRC and MA, LA had a closer relationship with sexual reproduction and spore development.

### 3.3. Expression Patterns of Key Bibenzyl Biosysthesis Genes in Four Parts of M. polymorpha

Stilbenecarboxylate synthase (STCS1) and 4-coumarate: Coenzyme A ligase (4CL) have proven to be crucial metabolic enzymes in the bibenzyl biosynthetic pathway [5]. To characterize the spatial signatures of bibenzyl synthesis-related enzymes, we determined the specific expression patterns of STCS1 and 4CL in the four parts of *M. polymorpha* by RT-Q-PCR analysis (Figure 3). Among them, MpSTCS1A was a reported gene sequence with STCS1 function [5], and Mp4CL was obtained by analyzing the genome database of *M. polymorpha*. The results showed that MpSTCS1A was highly expressed in the female *M. polymorpha* (ArM and TFM), and the expression level in ArM was about 4 times higher than that in AnM (Figure 3A). In addition, Mp4CL was mainly expressed in ArM, which showed a nearly 10-fold higher expression compared to other tissues (Figure 3B). The expression patterns of MpSTCS1A and Mp4CL were similar to the spatial distribution of bibenzyls in different tissues (Figure 2), suggesting the special features of bibenzyl synthesis-related enzymes can be reflected by the spatial distribution of bibenzyls.

### 3.4. Antioxidant Activity Evaluation of Four Parts from M. polymorpha

The antioxidant activities of ArM, AnM, TFM, and TMM were preliminary evaluated by TPC levels and presented by GAE mg/g FW [15,18]. As shown in Figure 4A, the content of total phenolic acid in ArM (2.61 ± 0.16 GAE mg/g FW) was significantly higher than that in male *M. polymorpha* (AnM and TMM, 1.45 ± 0.05 GAE mg/g FW and 2.10 ± 0.09 GAE mg/g FW, respectively), and AnM presented as the lowest TPC level.

In succession, three in vitro antioxidant activity assays, including DPPH radical scavenging ability, ABTS radical scavenging activity, and ferric reducing antioxidant power, were tested and expressed by the Trolox equivalent antioxidant capacity (TE μg/g FW), as shown in Figure 4B–D. These assays had similar trends and showed a certain correlation between TPC levels. The antioxidant activities of the extracts from the female *M. polymorpha* (ArM and TFM) were significantly higher than that of the male (AnM and TMM). Compared with ArM, TFM, and TMM, AnM had the lowest antioxidant activity for the DPPH method (Figure 1B). The most significant ABTS radical scavenging activity was exhibited by the ArM extract, which was about two times as much as that of the lowest value observed for the AnM extract, followed by TFM and TMM (Figure 4C). For the FRAP method, it had the same trend as the ABTS method, but there were no significant differences in antioxidant capacity between ArM and TFM (Figure 4D).

In this study, TPC, antioxidant capacity, spatio-chemical information, and related gene expression levels of four parts from *M. polymorpha* were positively correlated, which helped to visualize antioxidant heterogeneity and provides hints at photoprotective potentials of bibenzyls during sexual reproduction.

### 3.5. Analysis of Gene Expression Patterns and TPC Responses after UV Treatment

To further interrogate the gene and component responses to UV-B stress, we analyzed the gene expression patterns of the MpSTCS1A via RT-Q-RCR in response to UV-B for 3 h at different time points (0, 6, 12, 24, 36, 48, and 60 h) in the male and female *M. polymorpha* thalli, respectively. The results indicated that the MpSTCS1A transcripts in TFM were induced by UV-B irradiation with a more than 100-fold increase at 12 h. The relative expression level of MpSTCS1A decreased sharply at 24 h and increased again at 36 h. The same pattern was also exhibited at subsequent time points, suggesting that there may be a circadian rhythm in the relative expression levels of MpSTCS1A (Figure 5A). In TMM, the relative expression level of MpSTCS1A was lower compared with TFM, but it showed the same trend as TFM (Figure 5B). The TPC in TFM and TMM were at a lower level within 12 h of UV-B irradiation compared with the control group, but the content of total phenolic acid remained at a higher level after 12 h (Figure 5C,D). These data confirmed the positive role in liverwort UV-B protection of STCS1 and bibenzyls. Compared to gene response, the delay of component response was probably due to the time consumption of biosynthesis.

### 3.6. Promotion Effect of LA and Extraction of ArM on Spore Germination under Normal Conditions and UV Stress

Based on the above results, we speculated that bibenzyls, especially LA, could play a vital role in spore propagation. In this study, we explored the effect of LA and extraction of ArM on spore germination under normal conditions and UV stress. The results showed that low concentrations of ArM extract and LA promoted spore germination and high concentrations inhibited it under normal conditions. The maximum promoting concentration is 6.25 μg/mL and 2.5 μg/mL, respectively. After that, the promoting effect gradually weakened until the inhibitory effect was exhibited with the increasing concentrations of ArM extracts and LA (Figure 6A,B). Compared with the control group, the lengths of germination tubes were obviously longer in the LA and ArM group, and the protonema was first generated under normal conditions with 2.5 μg/mL LA (Figure 6C).

The previous study indicated that LA had abscisic acid (ABA)-like activity in higher plants, and it could inhibit the germination of a variety of higher plant seeds [19]. Our work demonstrates that LA can accelerate spore germination and protonema development under abiotic stress, besides allelopathy under biotic stress.

## 4. Conclusions

As a model species of basal land plants, *M. polymorpha* has gained increasing attention in the investigation of metabolites and their corresponding biosynthesis pathways [5,20,21], but in situ metabolic profile is lacking. In this study, spatial distributions of bibenzyls in *M. polymorpha* were mapped using AFADESI-MSI in the negative ionization. Bibenzyls were discovered to be especially enriched in the archegoniophore, where LA was mainly distributed in the central region and its content varied along the maturity level of sporangia, which suggested stimulatory effects in the development of sporophytes, while other bisbibenzyls such as IRC and MA had stronger ions intensity in the periphery, which might be related to their antifeedant, antimicrobial, and allelopathy effects.

The results of gene expression, antioxidant activity, and spore germination were consistent with MSI. The expression of the key genes involved in bibenzyl biosysthesis and antioxidant activity in archegoniophore were both found to be higher than in other areas of *M. polymorpha*. Moreover, the expression level of MpSTCS1A and the content of total phenolic acid increased under UV-B stress. Meanwhile, a certain concentration of LA and extract of archegoniophore could stimulate the spore germination of *M. polymorpha*. Our work exhibited the positive role of bibenzyls from the archegoniophore of *M. polymorpha* in UV-B protection and spore propagation.

The combination of spatial metabolic profile and region-specific antioxidant activity evaluation provides new insights into the understanding of stress tolerance and spore propagative protection of liverworts.

## Figures and Tables

**Figure 1 antioxidants-11-02157-f001:**
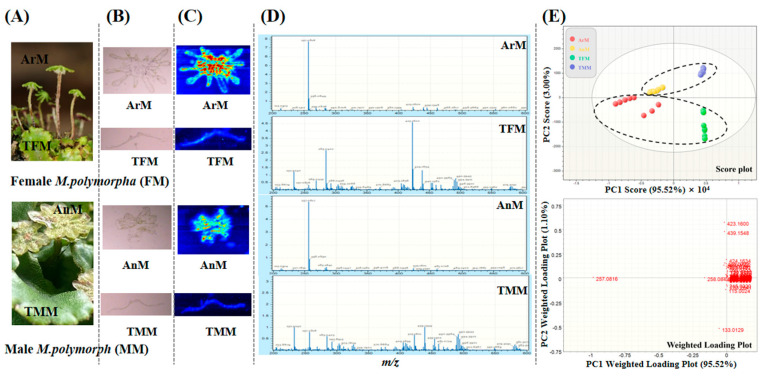
Multivariate statistical analysis for characteristic metabolites exploration of *M. polymorpha* using AFADESI. (**A**) Optical images of different plant parts of female and male *M. polymorpha*. (**B**) Optical images of cross sections of four parts (ArM, TFM, AnM, and TMM). (**C**) Negative mode AFADESI images of ArM, TFM, AnM, and TMM with the mass ranging from m/z 200 to 600. (**D**) Negative mode AFADESI mass spectra from ArM, TFM, AnM, and TMM. (**E**) PCA analysis of four parts from *M. polymorpha*.

**Figure 2 antioxidants-11-02157-f002:**
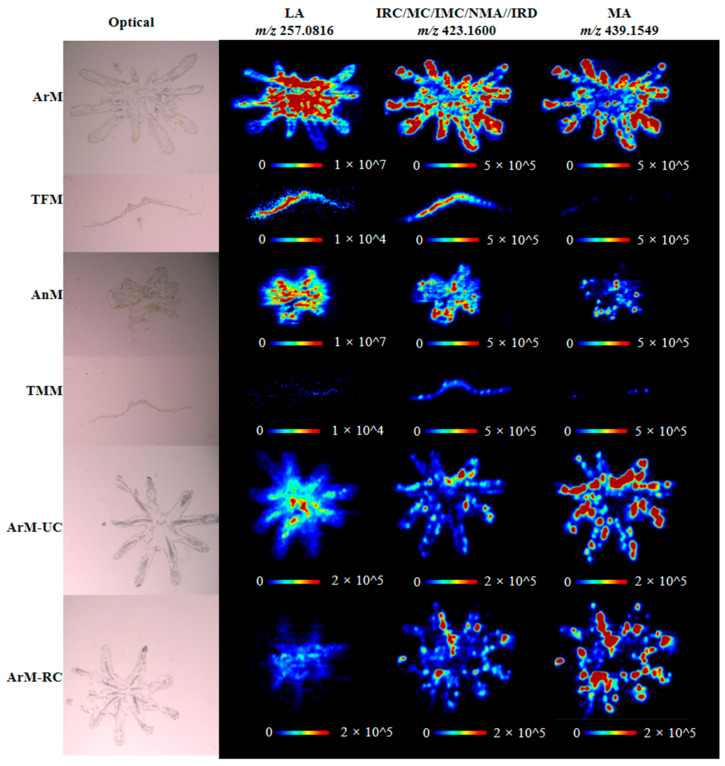
Spatial distribution of LA (*m/z* 257.0816), IRC/MC/IMC/NMA/IRD (*m/z* 423.1600), and MA (*m/z* 439.1549) in *M. polymorpha* receptacles and thalli based on AFADESI-MSI data.

**Figure 3 antioxidants-11-02157-f003:**
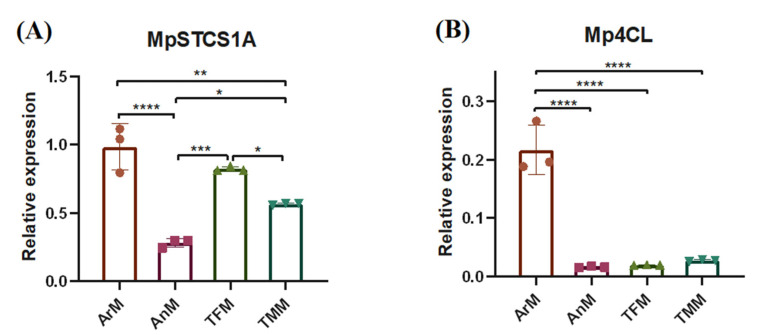
Expression patterns analysis of key genes involved in bibenzyl biosysthesis in four parts of *M. polymorpha*. (**A**) The expression level of MpSTCS1A in four tissues of *M. polymorpha*. (**B**) The expression level of Mp4CL in four tissues of *M. polymorpha*. Values are presented as mean ± standard deviation of three replications. Analysis of variance (ANOVA) and multiple range test (Tukey’s test) indicated significant differences. * *p* < 0.05, ** *p* < 0.01, *** *p* < 0.001, **** *p* < 0.0001.

**Figure 4 antioxidants-11-02157-f004:**
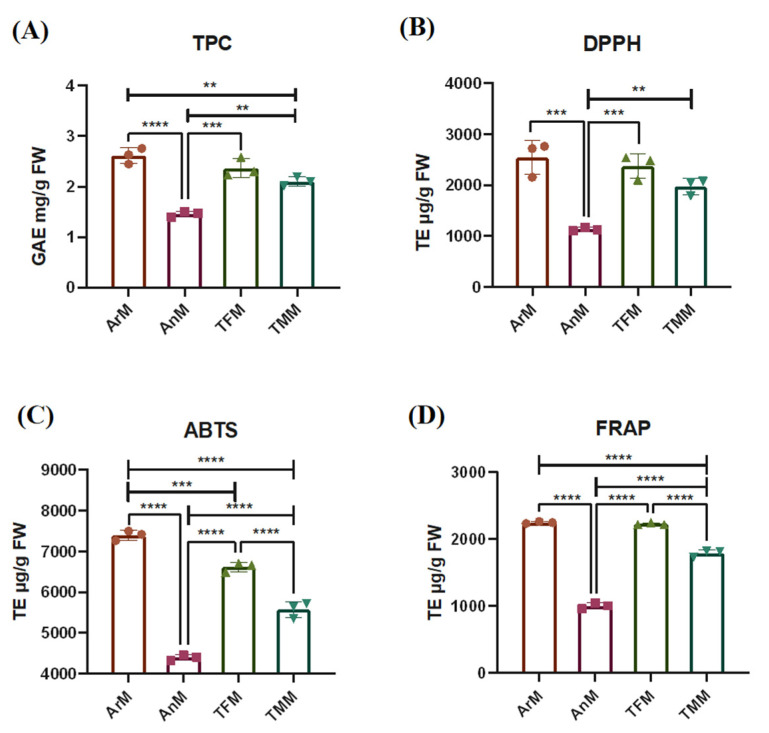
Total phenolic acids content and antioxidant activities of four parts from *M. polymorpha*. (**A**) Total phenolic acids content (TPC); (**B**) 2,2-diphenyl-1-picrylhydrazyl radical scavenging ability (DPPH); (**C**) 2,2′-azino-bis(3-ethylbenzothiazoline-6-sulfonic) acid radical scavenging activity (ABTS); (**D**) Ferric reducing antioxidant power (FRAP). GAE: gallic acid equivalent; TE: Trolox equivalent; FW: fresh weight. Values were presented as mean ± standard deviation of three replications. Analysis of variance (ANOVA) and multiple range test (Tukey’s test) indicated significant differences. ** *p* < 0.01, *** *p* < 0.001, **** *p* < 0.0001.

**Figure 5 antioxidants-11-02157-f005:**
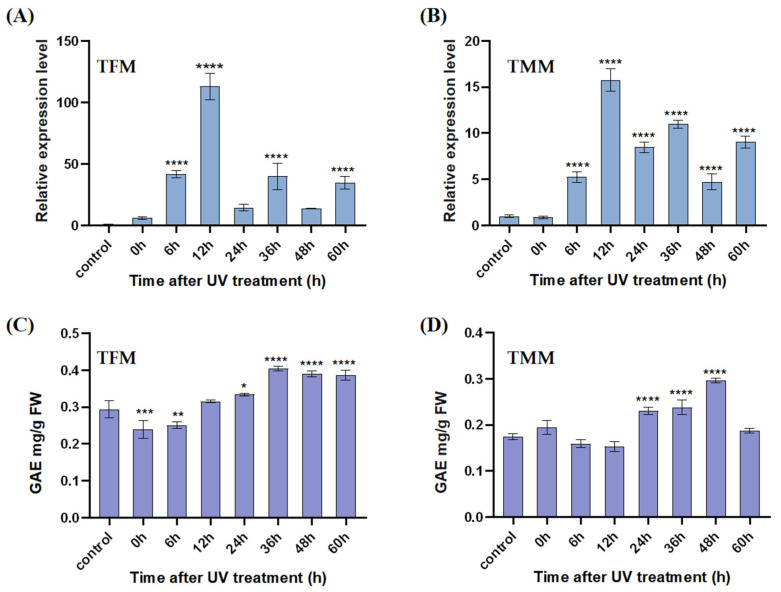
Expression patterns of MpSTCS1A in the female (**A**) and male (**B**) thallus of *M. polymorpha* and total phenolic acid content in the female (**C**) and male (**D**) thallus of *M. polymorpha* response to UV-B stress. The results are shown as the means of three replications with the standard deviations. Analysis of variance (ANOVA) and multiple range test (Tukey’s test) indicated significant differences. * *p* < 0.05, ** *p* < 0.01, *** *p* < 0.001, **** *p* < 0.0001. The total phenolic acid content in the extract was measured by the standard curve (y = 0.0483*x* + 0.0169, R^2^ = 0.9981).

**Figure 6 antioxidants-11-02157-f006:**
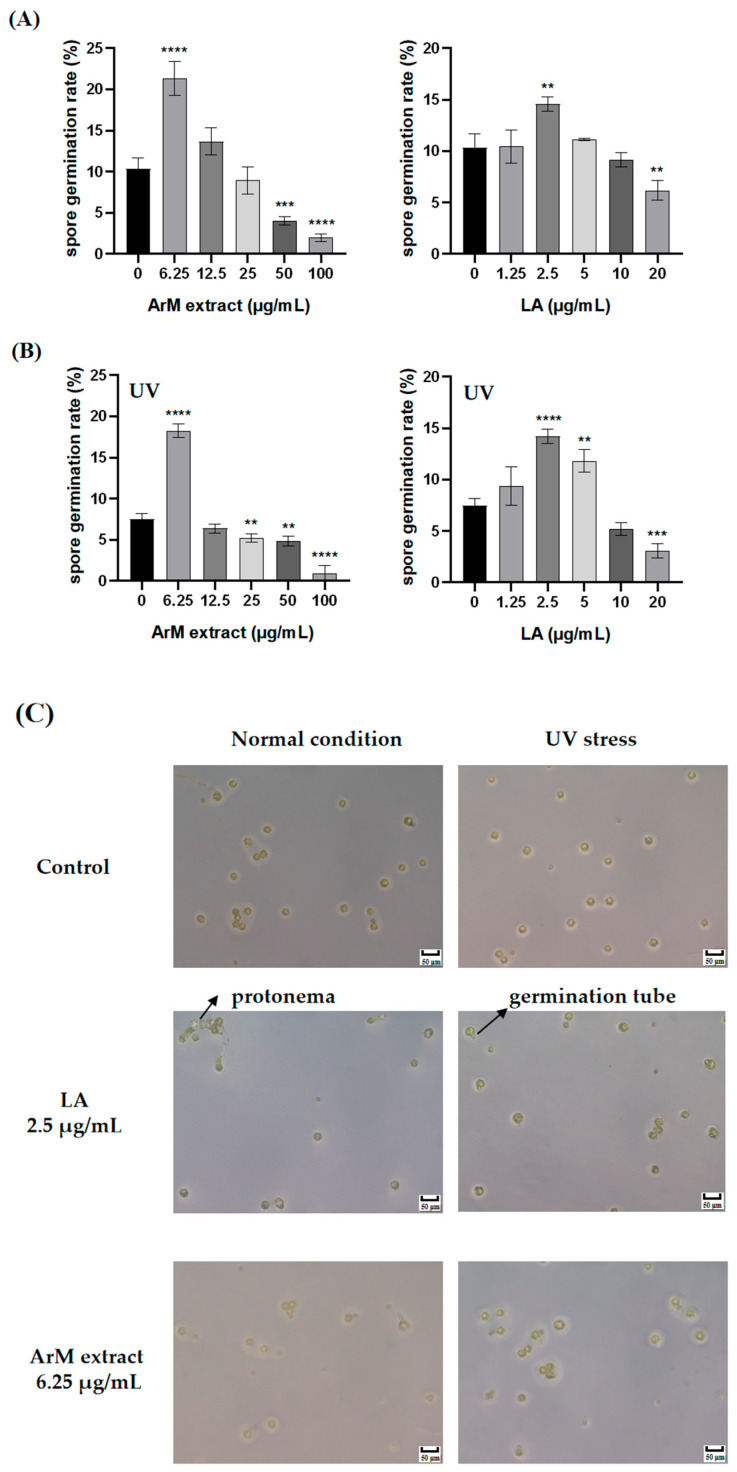
The effect of LA and extraction of ArM on spore germination under normal conditions and UV stress. (**A**) The spore germination rate of *M. polymorpha* treated with different concentrations of LA and ArM extract under normal conditions. (**B**) The spore germination rate of *M. polymorpha* treated with different concentrations of LA and ArM extract under UV stress. (**C**) Spore morphology of *M. polymorpha* treated with LA and ArM extract at the maximum promoting concentration under normal conditions and UV stress. Germination rates are expressed as means with the standard deviations. Three groups were set for each concentration, and the number of spores in each group was more than 50. Analysis of variance (ANOVA) and multiple range test (Tukey’s test) indicated significant differences. ** *p* < 0.01, *** *p* < 0.001, **** *p* < 0.0001.

## Data Availability

Data are contained within the article and Appendix A.

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
