# Peer review of "Spatial Distribution, Antioxidant Capacity, and Spore Germination-Promoting Effect of Bibenzyls from Marchantia polymorpha"

_antioxidants, 2022, doi:10.3390/antiox11112157_

Round 1
Reviewer 1 Report
The publication being submitted for evaluation is interesting and of good quality. It is centered on the spatial distribution, antioxidant capacity, and spore germination-promoting effect of bibenzyls from the representative species of liverworts. This species of liverworts is known for its ability to tolerate high concentrations of lead and other heavy metals in soil.
Obtained results are well documented. What is new is the visualization of the spatial distribution of metabolites In M. polymorpha at the reproductive stage.
In my opinion, it would be good to include information on the statistical analysis in the experimental part. Such information appears only under charts 3-5.
Author Response
Point 1:In my opinion, it would be good to include information on the statistical analysis in the experimental part. Such information appears only under charts 3-5.
Response 1: We appreciate the reviewer’s comment. 2.8. Statistical Analysis has been added in the experimental part of revised manuscript.
Reviewer 2 Report
The authors are dealing with a very interesting subject, worth further investigation.
There are just a few remarks from my side.
Please carefully check whether the latin name of the species is in italics in the entire manuscript.
In the M&M section, please add all volumes and concetrations of used reagenrs, for example, sodium carbonate does not have all this informaitons.
There should be space between numerical values and units.
Minutes and hours should be min and h.
Figure 6 should be revised, protonema is underlined.
In vitro, in vivo, in situ, all latin phrases should be in italics.
Author Response
Point 1: Please carefully check whether the latin name of the species is in italics in the entire manuscript.
Response 1: We appreciate the reviewer’s comment. We have checked and corrected this issue in the revised manuscript.
Point 2: In the M&M section, please add all volumes and concetrations of used reagenrs, for example, sodium carbonate does not have all this informaitons.
Response 2: We agree with the reviewer’s comment. The volume of sodium carbonate and the concentration of DPPH solution have been added in the revised manuscript.
Point 3: There should be space between numerical values and units.
Response 3: We have re-checked the entire manuscript, and added spaces between values and units.
Point 4: Minutes and hours should be min and h.
Response 4: We appreciate the reviewer’s comment. We have corrected this issue in the revised manuscript.
Point 5: Figure 6 should be revised, protonema is underlined.
Response 5: Figure 6 has been revised, and the underline has been deleted in the revised manuscript.
Point 6: In vitro, in vivo, in situ, all latin phrases should be in italics.
Response 6: “In vitro” and "in situ" has been changed to “in vitro” and "in situ" in the revised manuscript.
Reviewer 3 Report
Marchantia polymorpha produces bibenzyls and bisbibenzyls as prominent secondary metabolites. Their pathway resembles for its beginning that of flavonoids in vascular plants (including the enzyme 4CL and a form of PKS). Zhang and coauthors show that these compounds act as antioxidants and respond to UV-B stress. A technically very interesting part is the in situ MS exploration of where exactly in the Marchantia tissues different metabolites reside.
The paper is easy to read and gives important background information on liverwort secondary metabolism. Two minor comments:
1. In the UV-B treatment (Mat&Meth) it is not clear how the experiment was done. If the times are correct, the sampling points follow a 3 hour exposure. Are time points counted from the beginning of the exposure, or after a 3 hour exposure? Or should 3h be 3d and sampling was done during the UV-B exposure?
2. The software called “Compound Discoverer” lacks a reference.
Author Response
Point 1: In the UV-B treatment (Mat&Meth) it is not clear how the experiment was done. If the times are correct, the sampling points follow a 3 hour exposure. Are time points counted from the beginning of the exposure, or after a 3 hour exposure? Or should 3h be 3d and sampling was done during the UV-B exposure?
Response 1: We appreciate the reviewer’s comment. in our study. Sampling time points were counted after 3 h exposure. We have clarified this issue in the revised manuscript.
Point 2: The software called “Compound Discoverer” lacks a reference.
Response 2: Reference [17]: "Cerrato, A.; Aita, S.E.; Capriotti, A.L.; Cavaliere, C.; Montone, C.M.; Piovesana, S.; Laganà, A. Fully automatized detection of phosphocholine-containing lipids through an isotopically labeled buffer modification workflow. Chem. 2021, 93, 15042−15048." has been added in the revised manuscript.